# New Resistor-Less Electronically Controllable ±C Simulator Employing VCII, DVCC, and a Grounded Capacitor

**Giuseppe Ferri** [1], **Leila Safari** [1,2], **Gianluca Barile** [1,2,*], **Massimo Scarsella** [1] and **Vincenzo Stornelli** [1,2]

1   Department of Industrial and Information Engineering (DIIIE), University of L'Aquila, 67100 L'Aquila, Italy; giuseppe.ferri@univaq.it (G.F.); leilasafari@yahoo.com (L.S.); massimo.scarsella@graduate.univaq.it (M.S.); vincenzo.stornelli@univaq.it (V.S.)
2   DEWS, University of L'Aquila, 67100 L'Aquila, Italy
*   Correspondence: gianluca.barile@univaq.it

**Abstract:** In this paper, a new realization of electronically controllable positive and negative floating capacitor multiplier (±C) is presented. The peculiarity of the presented topology is that, for the first time, it implements a floating equivalent capacitor between its two input terminals, rather than a grounded one. To achieve the best performance, we simultaneously use the advantages provided by the current conveyor and its dual circuit, the voltage conveyor. The proposed topology is resistor free and employs one dual-output second-generation voltage conveyor (VCII$^{\pm}$) and one electronically tunable differential voltage current conveyor (E-DVCC) as active building blocks (ABBs) and a single grounded capacitor. The value of the simulated capacitor is controlled by means of a control voltage $V_C$ which is used to control the current gain between *X* and *Z* terminals of E-DVCC. The circuit is free from any matching condition. A complete non-ideal analysis by considering parasitic impedances as well as non-ideal current and voltage gains of the used ABBs is presented. The proposed circuit is designed at the transistor level in 0.18 μm and ±0.9 V supply voltage. Simulation results using the SPICE program show a multiplication factor ranging from ±10 to ±25.4 with a maximum error of 0.56%. As an example, the application of the achieved floating capacitor as a standard high pass filter is also included.

**Keywords:** VCII; floating capacitance multiplier; DVCC; low voltage; electronically tunable; high pass filter

## 1. Introduction

The advantages of capacitance multipliers in realizing large value capacitors in the CMOS process regarding the cost and chip size reduction are well known [1–6]. Capacitance multipliers find wide use in applications requiring large time constants, such as low-frequency filters in biomedical applications. In recent years, capacitor multiplier implementation has become a more important topic and several techniques have been developed to realize capacitor multipliers [7–19]. The feature of electronic tuning is considered a great advantage because it gives suitable tuning of filters, oscillators, and other circuits employing the simulated capacitor. Based on the intended application, simulated capacitors can be either grounded or floating.

Literature survey shows that various active building blocks (ABBs) have been used to develop floating capacitance multipliers [7–21]. The used ABBs are mainly the second-generation current conveyor (CCII) [22,23], current-controlled differential difference current conveyor (CCDDCC) [24,25], current follower transconductance amplifier (CFTA) [26], voltage current gain controlled second-generation current conveyor (VGC-CCII), dual output second-generation current conveyor (Do-CCII) [27], current-controlled second-generation current conveyor (CCCII), differential voltage current conveyor (DVCC), multiple output DVCC(MODVCC), differential voltage current conveyor transconductance amplifier

(DVCCTA), operational transconductance amplifier (OTA), and flipped voltage follower (FVF) [28]. However, the previous implementations of floating capacitors in [7–19] suffer from several drawbacks. For example, the circuit reported in [7] employs two OTA, one CCII, and one resistor, and the achieved floating capacitor shows a high error. The power consumption of the circuits reported in [9,11–13,17] is high. The solution reported in [9] uses five CFTA and the one reported in [19] requires four OTA. The circuits reported in [12,13,15,17,18,20,21] lack electronic tuning capability. In the circuits reported in [10,14,17,18,21], a floating capacitor is employed. More importantly, in [17], two floating capacitors are used which must be well matched. Even worse, in [18], four floating capacitors are used, and matching is required between these four capacitors.

Recently, the application of a second-generation voltage conveyor (VCII) [29–31] as the dual circuit of CCII has been investigated in realizing impedance simulators [2,32–34]. In particular, in [2,33], its application in simulating grounded capacitor multiplier is shown, where the VCII-based grounded capacitors outperform the previously reported works in many terms.

In this paper, we intend to use the intrinsic capabilities of VCII in designing high-performance floating capacitor multipliers. To this aim, we introduce a new topology to implement a floating capacitor using a dual output VCII, an electronically controllable DVCC (which we call E-DVCC), and a grounded capacitor. In order to have an electronic tuning capability, the current gain between the $X$ and $Z$ terminals of E-DVCC is controlled using a control voltage, $V_C$. Employing both VCII and DVCC in the proposed topology, the benefits of voltage conveyors and current conveyors are combined to achieve the desired properties of floating capacitance multiplication, low-voltage low-power operations, reduced associated parasitic elements, electronic tunability, low error, no matching condition, and simple circuitry. Both positive and negative multiplications can be implemented by the proposed circuit. A complete non-ideal analysis is given, and the SPICE simulation results are reported. The application of the proposed floating capacitor as a standard high pass filter is also given, as an application of study.

The organization of this paper is as follows: in Section 2, the proposed circuit is introduced; in Section 3, non-ideal analysis is performed; Section 4 shows the implementation of the active blocks; Section 5 includes the simulation results; and finally, Section 6 concludes the paper.

## 2. The Proposed VCII-Based Floating ±C Multiplier

The proposed VCII-based floating +C multiplier is shown in Figure 1. It is composed of one VCII$^\pm$, one E-DVCC, and a single grounded capacitor. Here, we use the intrinsic potential of voltage conveyors and current conveyors to achieve a high-performance floating capacitor multiplier. In the E-DVCC, the current gain between the $X$ and $Z$ terminals can be regulated by the voltage $V_C$ which allows us to electronically control the value of simulated +C. Consequently, as it will be shown, the value of the simulated floating capacitor can be varied by $V_C$.

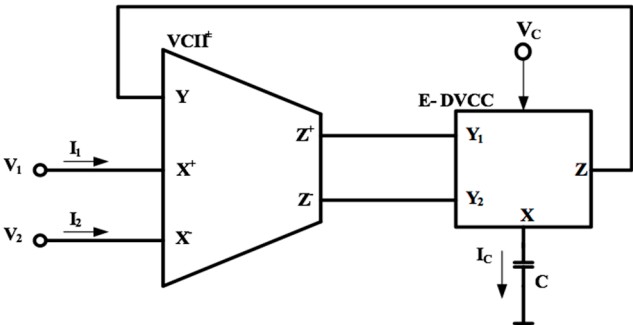

**Figure 1.** The proposed floating positive capacitance multiplier.

In the following, we perform the analysis of the proposed circuit: operation the VCII$^{\pm}$ is ideally expressed by the equations:

$$\begin{bmatrix} I_{X+} \\ I_{X-} \\ V_{Z+} \\ V_{Z-} \\ V_Y \end{bmatrix} = \begin{bmatrix} 1 & 0 & 0 & 0 \\ -1 & 0 & 0 & 0 \\ 0 & 1 & 0 & 0 \\ 0 & 0 & 1 & 0 \\ 0 & 0 & 0 & 0 \end{bmatrix} \begin{bmatrix} I_Y \\ V_{X+} \\ V_{X-} \\ I_Z \end{bmatrix} \tag{1}$$

while the following matrix represents the ideal operation of E-DVCC:

$$\begin{bmatrix} V_X \\ I_{Y1} \\ I_{Y2} \\ I_Z \end{bmatrix} = \begin{bmatrix} 1 & -1 & 0 & 0 \\ 0 & 0 & 0 & 0 \\ 0 & 0 & 0 & 0 \\ 0 & 0 & K & 0 \end{bmatrix} \begin{bmatrix} V_{Y1} \\ V_{Y2} \\ I_X \\ V_Z \end{bmatrix} \tag{2}$$

Being $K$, the tunable current gain between the $X$ and $Z$ terminals. Using Equation (3a,b) for $V_{Z+}$ and $V_{Z-}$ we have:

$$V_{Z+} = V_1, \tag{3a}$$

$$V_{Z-} = V_2, \tag{3b}$$

where $V_1$ and $V_2$ are the input voltages. Using (2) and (3a,b) the voltage across $C$ is:

$$V_X = V_{Y1} - V_{Y2} = V_1 - V_2 \tag{4}$$

From Equation (4), the produced current across $C$ is found as:

$$I_C = sCV_X = sC(V_1 - V_2) \tag{5}$$

The current produced across $C$ is conveyed to the E-DVCC $Z$ terminal while it is amplified by the gain of $K$ as follows:

$$I_Z = KI_C = sKC(V_1 - V_2) \tag{6}$$

As the $Z$ terminal of E-DVCC is directly connected to the $Y$ terminal of VCII$^{\pm}$, we have $I_Y = I_Z$. According to Equation (1), the current at $Y$ terminal of VCII$^{\pm}$ is conveyed to the $X+$ and $X-$ terminals by gains of $(+1)$ and $(-1)$, respectively, so for $I_1$ and $I_2$ we have:

$$I_1 = I_{X+} = sKC(V_1 - V_2) \tag{7a}$$

$$I_2 = I_{X-} = -sKC(V_1 - V_2) \tag{7b}$$

From Equation (7a,b), the input impedance is found as:

$$Z_{in} = 1/sKC \tag{8}$$

From which the equivalent capacitor is ideally:

$$C_{eq} = KC \tag{9}$$

Figure 2 shows the realization of a floating negative capacitance simulator where, differently from Figure 1, $Z+$ and $Z-$ terminals of VCII$^{\pm}$ are respectively connected to $Y_2$ and $Y_1$ terminals of the E-DVCC. For Figure 2, it can be easily shown that the equivalent floating capacitor is ideally given

$$C_{eq} = -KC \tag{10}$$

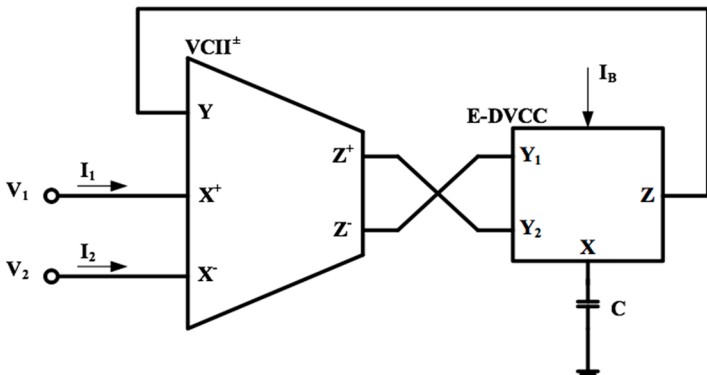

**Figure 2.** The proposed floating negative capacitance multiplier.

### 3. Non-Ideal Analysis

Figure 3 shows a more detailed representation of the used ABBs showing the parasitic impedances associated with each port of $VCII^{\pm}$ and E-DVCC. In non-ideal conditions, by considering these non-idealities, the operation of $VCII^{\pm}$ is expressed by the following matrix:

$$
\begin{bmatrix} I_{X+} \\ I_{X-} \\ V_{Z+} \\ V_{Z-} \\ V_Y \end{bmatrix} = \begin{bmatrix} \beta_{v1} & \frac{1}{r_{X+}} & 0 & 0 \\ -\beta_{v2} & 0 & \frac{1}{r_{X-}} & 0 \\ 0 & \alpha_{v1} & 0 & r_{Z+} \\ 0 & 0 & \alpha_{v2} & r_{Z-} \\ r_Y & 0 & 0 & 0 \end{bmatrix} \begin{bmatrix} I_Y \\ V_{X+} \\ V_{X-} \\ I_Z \end{bmatrix}
\tag{11}
$$

where $\beta_{v1}$ and $\beta_{v2}$ are current gains between the $Y$ and $X+$ terminals and between the $Y$ and $X-$ terminals, respectively, $\alpha_{v1}$ and $\alpha_{v2}$ are voltage gains between the $X+$ and $Z+$ terminals and between the $X-$ and $Z-$ terminals, respectively, with ideal values of unity. Parameters $r_Y$, $r_{X+}$, $r_{X-}$, $r_{Z+}$, and $r_{Z-}$ are the low-frequency impedances at the $Y$, $X+$, $X-$, $Z+$, and $Z-$ terminals, respectively. The ideal values of $r_Y$, $r_{X+}$, $r_{X-}$, $r_{Z+}$, and $r_{Z-}$ are zero, infinite, infinite, zero, and zero, respectively. Matrix Equation (12) represents the real operation of E-DVCC:

$$
\begin{bmatrix} V_X \\ I_{Y1} \\ I_{Y2} \\ I_Z \end{bmatrix} = \begin{bmatrix} \alpha_{c1} & -\alpha_{c2} & r_X & 0 \\ 0 & 0 & 0 & 0 \\ 0 & 0 & 0 & 0 \\ 0 & 0 & K & \frac{1}{r_Z} \end{bmatrix} \begin{bmatrix} V_{Y1} \\ V_{Y2} \\ I_X \\ V_Z \end{bmatrix}
\tag{12}
$$

where $\alpha_{c1}$ is voltage gain between the $Y_1$ and $X$ terminals and $\alpha_{c2}$ is voltage gain between $Y_2$ and $X$ terminals, $K$ is tunable current gain between the $X$ and $Z$ terminals, and $r_x$ and $r_z$ are low-frequency parasitic impedances at the $X$ and $Z$ terminals, respectively. The ideal value of $\alpha_{c1}$, $\alpha_{c2}$, $r_x$, and $r_z$ are unity, unity, zero, and infinite, respectively.

Figure 4 shows the proposed positive floating $C$ simulator while all low-frequency parasitic impedances are considered.

In Figure 4, for $V_{Z+}$ and $V_{Z-}$ we have:

$$
V_{Z+} = \alpha_{v1} V_1
\tag{13a}
$$

$$
V_{Z-} = \alpha_{v2} V_2
\tag{13b}
$$

Using Equation (13) and considering the fact that $r_{Y1} \gg r_{Z+}$ and $r_{Y2} \gg r_{Z-}$, for $V_{Y1}$ and $V_{Y2}$ we have:

$$
V_{Y1} = \frac{r_{Y1}}{r_{Z+} + r_{Y1}} V_{Z+} \approx \alpha_{v1} V_1
\tag{14a}
$$

$$
V_{Y2} = \frac{r_{Y2}}{r_{Z-} + r_{Y2}} V_{Z-} \approx \alpha_{v2} V_2
\tag{14b}
$$

Using Equations (12)–(14), $I_C$ is found as:

$$I_C = \frac{sC}{1 + sCr_X}(\alpha_{c1}\alpha_{v1}V_1 - \alpha_{c2}\alpha_{v2}V_2) \tag{15}$$

Assuming $r_Y << r_Z$, using Equations (12) and (15) we have:

$$I_Y = \frac{sCK}{1 + sCr_X}(\alpha_{c1}\alpha_{v1}V_1 - \alpha_{c2}\alpha_{v2}V_2) \tag{16}$$

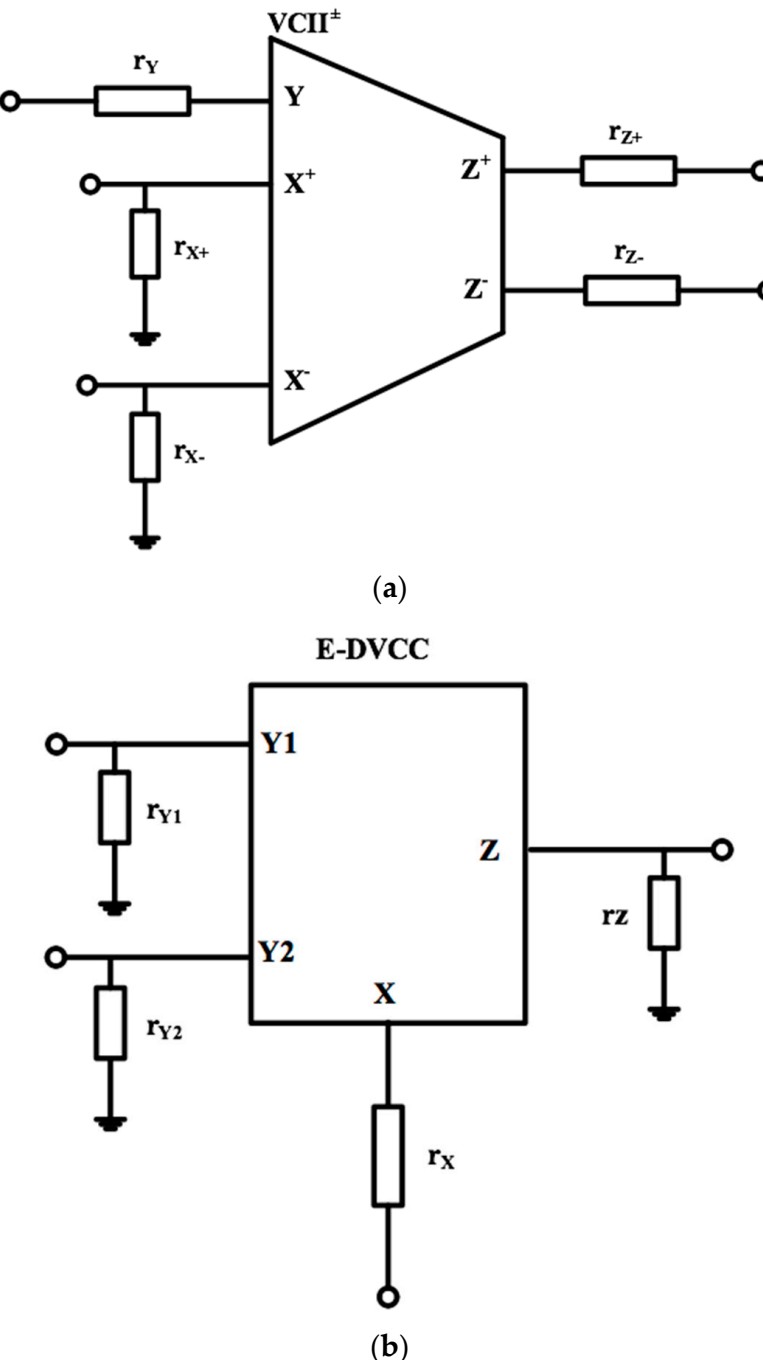

(a)

(b)

**Figure 3.** Non-ideal impedances of (**a**) VCII$^\pm$ and (**b**) E-DVCC.

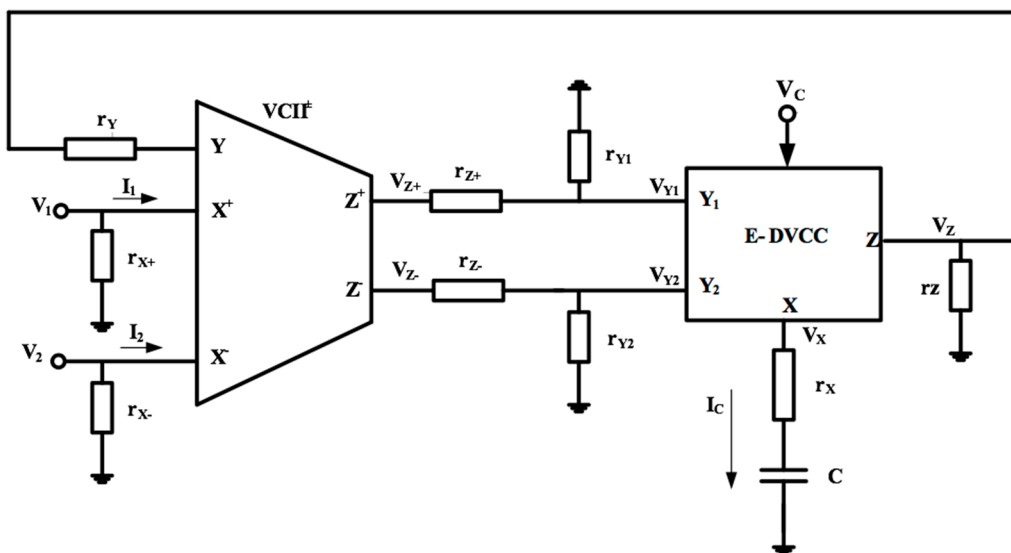

**Figure 4.** Proposed positive floating capacitor simulator with parasitic impedances.

According to Equation (11), neglecting the parasitic components (that is $1/r_{x\pm} \cong 0$), the current at Y terminal of VCII$^{\pm}$ is conveyed to the X+ and X− terminals by gains of $(+\beta_{v1})$ and $(-\beta_{v2})$, respectively, and for $I_1$ and $I_2$ we have:

$$I_1 = I_{X+} = \beta_{v1}I_Y = \frac{sCK}{1+sCr_X}(\alpha_{c1}\alpha_{v1}V_1 - \alpha_{c2}\alpha_{v2}V_2)\beta_{v1} \tag{17a}$$

$$I_2 = I_{X-} = -\beta_{v2}I_Y = -\frac{sCK}{1+sCr_X}(\alpha_{c1}\alpha_{v1}V_1 - \alpha_{c2}\alpha_{v2}V_2)\beta_{v2} \tag{17b}$$

In the worst case, $\beta_{v1}$ and $\beta_{v2}$ can be expressed as the following where $\varepsilon << 1$:

$$\beta_{v1} = (1 \pm \varepsilon) \tag{18a}$$

$$\beta_{v2} = (1 \mp \varepsilon) \tag{18b}$$

Inserting Equation (18a) into Equation (17a) and Equation (18b) into Equation (17b) gives:

$$I_1 = \frac{sCK}{1+sCr_X}(\alpha_{c1}\alpha_{v1}V_1 - \alpha_{c2}\alpha_{v2}V_2) \pm \frac{sCK}{1+sCr_X}(\alpha_{c1}\alpha_{v1}V_1 - \alpha_{c2}\alpha_{v2}V_2)\varepsilon \tag{19a}$$

$$I_2 = -\frac{sCK}{1+sCr_X}(\alpha_{c1}\alpha_{v1}V_1 - \alpha_{c2}\alpha_{v2}V_2) \mp \frac{sCK}{1+sCr_X}(\alpha_{c1}\alpha_{v1}V_1 - \alpha_{c2}\alpha_{v2}V_2)\varepsilon \tag{19b}$$

Using Equation (19), the input admittance of the proposed circuit is found as:

$$Z_{in} = \frac{(V_1 - V_2)(1+sCr_X)}{sCK(\alpha_{c1}\alpha_{v1}V_1 - \alpha_{c2}\alpha_{v2}V_2) \pm sCK(\alpha_{c1}\alpha_{v1}V_1 - \alpha_{c2}\alpha_{v2}V_2)\varepsilon} \tag{20}$$

The parameters $\alpha_{c1}$, $\alpha_{c2}$, $\alpha_{v1}$, and $\alpha_{v2}$ can be defined as Equation (21a–d) in which $\varepsilon_c << 1$ and $\varepsilon_v << 1$:

$$\alpha_{c1} = (1 \pm \varepsilon_c) \tag{21a}$$

$$\alpha_{c2} = (1 \mp \varepsilon_c) \tag{21b}$$

$$\alpha_{v1} = (1 \pm \varepsilon_v) \tag{21c}$$

$$\alpha_{v2} = (1 \mp \varepsilon_v) \tag{21d}$$

Inserting Equation (21a–d) into Equation (20) and simplifying the equation gives:

$$Z_{in} = \frac{1 + sCr_X}{sCK[1 \pm \varepsilon_c \pm \varepsilon_v]} = \frac{1}{sCK[1 \pm \varepsilon_c \pm \varepsilon_v]} + \frac{r_X}{K[1 \pm \varepsilon_c \pm \varepsilon_v]} \tag{22}$$

From Equation (22), the value of $C_{eq}$ is found as:

$$C_{eq} = CK[1 \pm \varepsilon_c \pm \varepsilon_v] \tag{23}$$

As can be seen, the non-ideal gains have a negligible effect on the value of $C_{eq}$, and in addition, using $K$, the value of $C_{eq}$ is adjustable.

The value of the series of the parasitic capacitance of the simulated capacitor is found as:

$$R_{seri} = \frac{r_X}{K[1 \pm \varepsilon_c \pm \varepsilon_v]} \tag{24}$$

As can be seen from Equation (24), the value of $R_{seri}$ is reduced by increasing multiplication factor $K$. The equivalent circuit of the proposed capacitor multiplier is shown in Figure 5 where $C_{eq}$ is the proposed electronically tunable equivalent floating capacitance. $r_{x\pm}$ is the parasitic parallel resistances of the $X\pm$ terminals, and $R_{seri}$ is the parasitic series resistance of the simulated capacitor. A similar analysis can be performed for negative floating $C$.

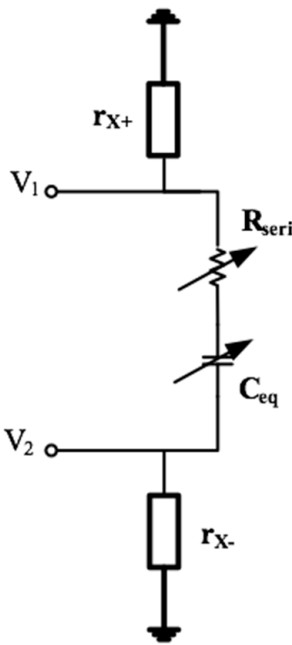

**Figure 5.** The equivalent circuit of the proposed floating C multiplier.

## 4. The CMOS Implementation of VCII$^{\pm}$ and E-DVCC

A realization of the internal structure of VCII$^{\pm}$ is shown in Figure 6. The low impedance at the $Y$ port is provided by the negative feedback loop made by the $M_1$–$M_5$ transistors. The input current to the $Y$ port is transferred to the $X+$ port by a simple current mirror $M_6$–$M_7$. The voltage produced at the $X+$ node is transferred to the $Z+$ port by voltage buffer made by $M_8$–$M_{12}$. Similarly, the $Y$ port input current is reversed by the current mirror, $M_{14}$–$M_{15}$, and transferred to the $X-$ port. The voltage at the $X-$ port is transferred to the $Z_-$ port by $M_{16}$–$M_{20}$. The transistor-level implementation of an electronically tunable E-DVCC is shown in Figure 7. The difference between input voltages $V_1$ and $V_2$ is produced at low impedance $X$ port utilizing the negative feedback loop established by differential pairs, $M_1$–$M_2$ and $M_3$–$M_4$, along with $M_5$–$M_7$ transistors as reported in [27]. In the previously reported works, DVCC is used while the gain between the $X$ and $Z$ terminals

is unity. Here, we add a gain cell between the X and Z terminals to electronically tune the current at the Z terminal. The used gain cell is a variable gain current mirror previously reported in [35].

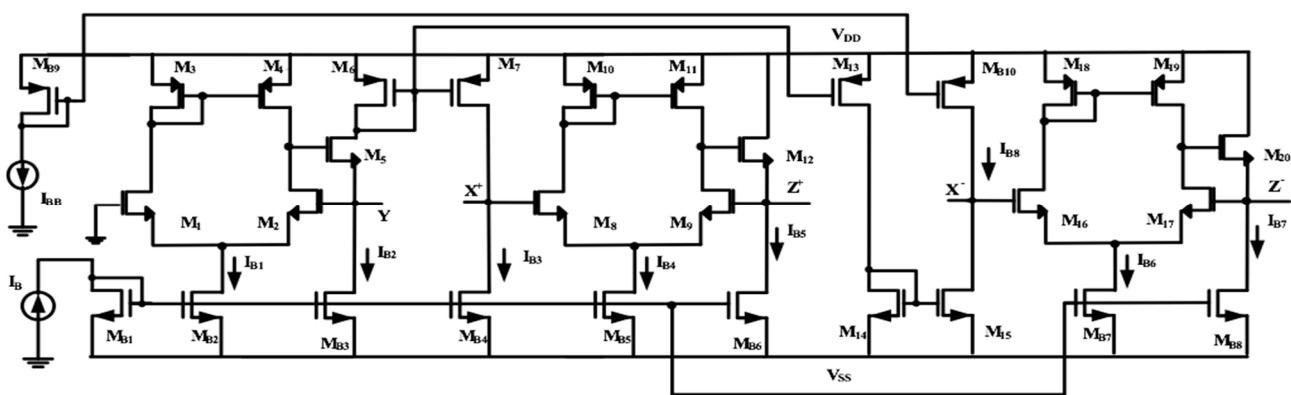

**Figure 6.** CMOS implementation of VCII$^{\pm}$.

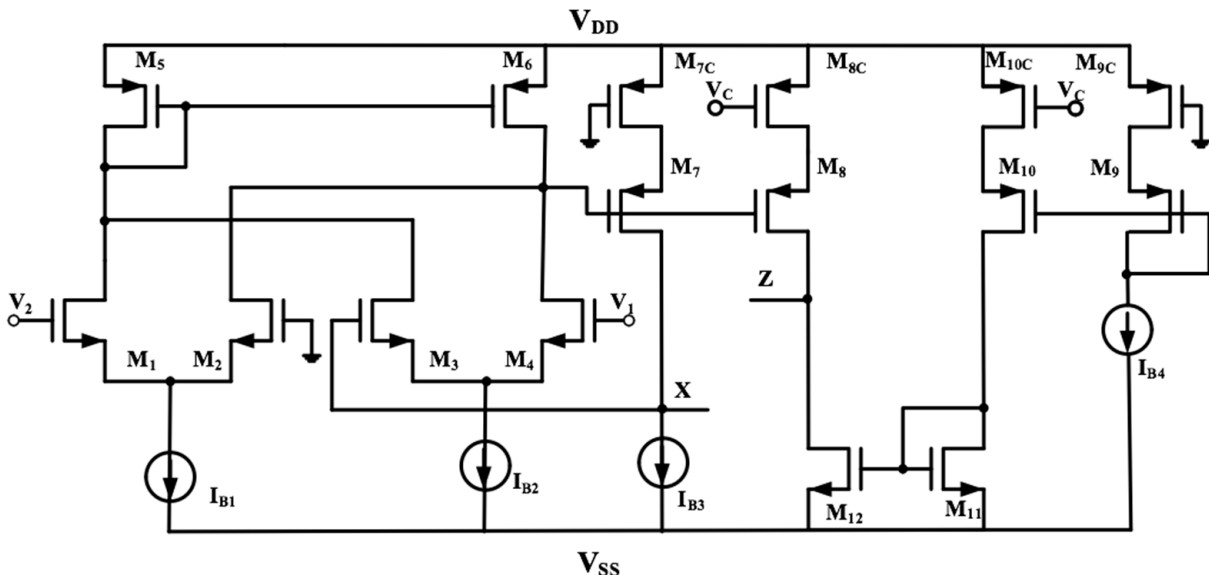

**Figure 7.** CMOS implementation of E-DVCC.

The current of the X terminal is transferred to the gain cell using the current mirror made of $M_7$–$M_{7C}$ and $M_8$–$M_{8C}$. As was described in [35], the current gain between the X and a Z terminal is tunable by control voltage $V_C$, expressed as:

$$K = \frac{I_Z}{I_X} = \frac{gm_7}{gm_8} \frac{1 + gm_8 rds_{M8c}}{1 + gm_7 rds_{M7c}} \tag{25}$$

being:

$$rds_{M8c} = \frac{1}{\mu C_{ox} \frac{W_{M8c}}{L_{M8c}} (V_{dd} - V_c - V_{THP})} \tag{26a}$$

and

$$rds_{M7c} = \frac{1}{\mu C_{ox} \frac{W_{M7c}}{L_{Mc}} (V_{dd} - V_{THP})} \tag{26b}$$

where, according to the usual meaning of symbols, $gm_i$ and $rds_i$ represent the transconductance and the drain-source resistance of the $i$th transistor, respectively, $\mu$ is the mobility of the carriers, $C_{ox}$ represents the capacitance per unit of area of the specific fabrication

technology, $W/L_i$ is the ratio between the width and the length of the ith transistor, and $V_{TH}$ is the threshold voltage that is a technology-dependent parameter.

As it is seen from (26a), the current gain is controllable by $V_C$.

## 5. Simulation Results

The proposed floating capacitor of Figure 1 is simulated in 0.18 μm CMOS technology and supply voltage of ±0.9 V using the SPICE program. The used aspect ratios are reported in Table 1. All current sources are implemented by simple current mirrors with aspect ratio of W = 9 μm and L = 0.72 μm with the values of $I_{B1} = I_{B2} = I_{B3} = 30$ μA, IB4 = 31.4 μA. The performance parameters of VCII and E-DVCC are reported in Table 2.

**Table 1.** The used transistors aspect ratios.

| | **Transistor** | **M$_1$–M$_4$** | **M$_5$–M$_6$, M$_7$, M$_{8C}$, M$_{10C}$, M$_9$** | **M$_{7C}$, M$_{9C}$** | **M$_{11}$, M$_{12}$** | **M$_8$, M$_{10}$** |
|---|---|---|---|---|---|---|
| E-DVCC | Aspect Ratio (W/L) | 9 μm/0.18 μm | 72 μm/0.9 μm | 7.2 μm/0.9 μm | 9 μm/0.9 μm | 720 μm/0.9 μm |
| VCII | **Transistor** | **M$_1$–M$_2$, M$_8$–M$_9$, M$_{17}$–M$_{18}$, M$_6$–M$_7$, M$_{13}$** | **M$_3$–M$_4$, M$_{10}$–M$_{11}$, M$_{18}$–M$_{19}$** | | **M$_5$, M$_{12}$, M$_{20}$** | |
| | Aspect Ratio (W/L) | 9 μm/0.72 μm | 36 μm/0.72 μm | | 8 μm/0.72 μm | |

**Table 2.** The simulated low-frequency characteristics of VCII$^\pm$ and E-DVCC.

| **VCII$^\pm$** | | **E-DVCC** | | |
|---|---|---|---|---|
| $r_Y$ | 38.6 Ω | $\alpha_{c1}$ | | 0.998 |
| $r_{X+}$ | 147.5 kΩ | $\alpha_{c2}$ | | 0.997 |
| $r_{X-}$ | 133.7 kΩ | $r_x$ | | 574 Ω |
| | | $r_{Y1}, r_{Y2}$ | | >2 GΩ |
| $r_{Z+}$ | 37 Ω | $r_Z$ | $V_C = 0$ V | 120 kΩ |
| | | | $V_C = -0.45$ V | 80 kΩ |
| | | | $V_C = -0.9$ V | 50 kΩ |
| $r_{Z-}$ | 37 Ω | $K$ | $V_C = 0$ V | 10 |
| $\beta_1$ (DC) | 1.023 | | $V_C = -0.45$ V | 19.5 |
| $\beta_2$ (DC) | 1.01 | | $V_C = -0.9$ V | 25.4 |
| $\alpha_{v1}$ (DC) | 0.983 | $Pd$ | $V_C = 0$ V | 1.43 mW |
| $\alpha_{v2}$ (DC) | 0.983 | | $V_C = 0$ V | 2.03 mW |
| $Pd$ | 0.804 mW | | $V_C = -0.9$ V | 2.38 mW |

The value of *C* is set at 10 pF. To verify the functionality of the proposed circuit, the plot of impedance magnitude and phase response of the simulated floating capacitor and the ideal one are shown in Figure 8. By changing the value of $V_C$ from −0.9 V to 0 V, different multiplication factors are set. The frequency operation range of the proposed circuit results is equal to about two decades (from 10 kHz to 1 MHz). The maximum error value for different control voltages is shown in Table 3. As can be seen, the maximum error value is only 0.56%. The application of the proposed floating capacitor as a high pass filter is shown in Figure 9. The frequency response of the high pass filter for different control voltages is shown in Figure 10. The −3 dB frequencies of the filter are 100 kHz, 130 kHz, and 230 kHz for $V_C$ = −0.9 V, −0.45 V, and 0 V, respectively.

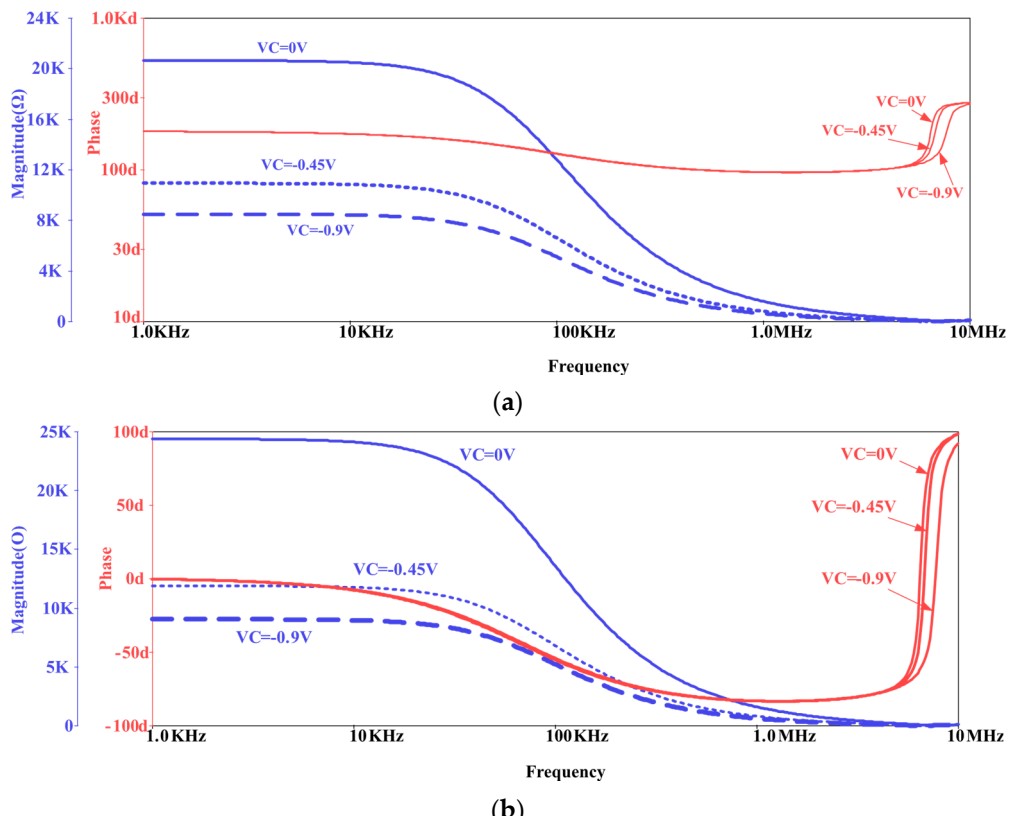

**Figure 8.** Frequency response of magnitude (blue) and phase (red) for the proposed (**a**) positive and (**b**) negative floating capacitor simulator.

**Table 3.** Deviation between theoretical and simulated values at different multiplication factors, $C = 10$ pF and f = 100 kHz.

| | $V_C$ | Multiplication Factor | Expected Value of $C_{eq}$ | Simulated Value of $C_{eq}$ | % Error |
|---|---|---|---|---|---|
| Positive simulator | $V_C = 0$ V | 10 | 100 pF | 99.4 pF | −0.6 |
| | $V_C = −0.45$ V | 19.5 | 195 pF | 196.1 pF | 0.56 |
| | $V_C = −0.9$ V | 25.4 | 254 pF | 255.4 pF | 0.55 |
| Negative simulator | $V_C = 0$ V | 10 | −100 pF | −99.47 pF | −0.53 |
| | $V_C = −0.45$ V | 19.5 | −195 pF | −196 pF | 0.51 |
| | $V_C = −0.9$ V | 25.4 | −254 pF | −255.3 pF | 0.51 |

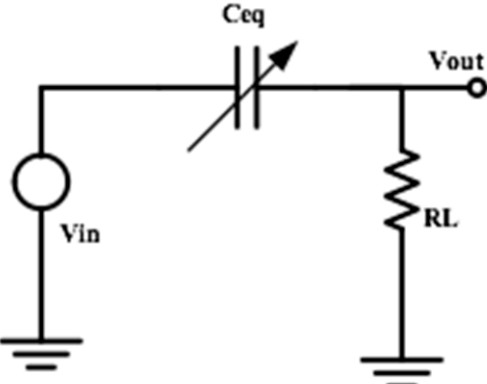

**Figure 9.** Application of the proposed simulated floating *C* as a high pass filter.

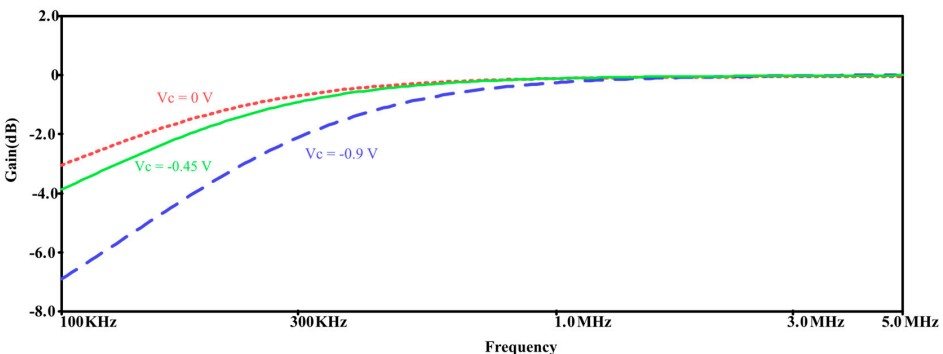

**Figure 10.** Frequency response of the high pass filter.

In order to examine the time domain response of the proposed circuit, an input voltage with a peak-to-peak value of 0.4 V and frequency of 1 MHz is applied. For different values of control voltages of 0, −0.45 V, and −0.9 V, the value of the input current phase is −101°, −101.4°, and −101°, respectively. The time domain responses are shown in Figure 11.

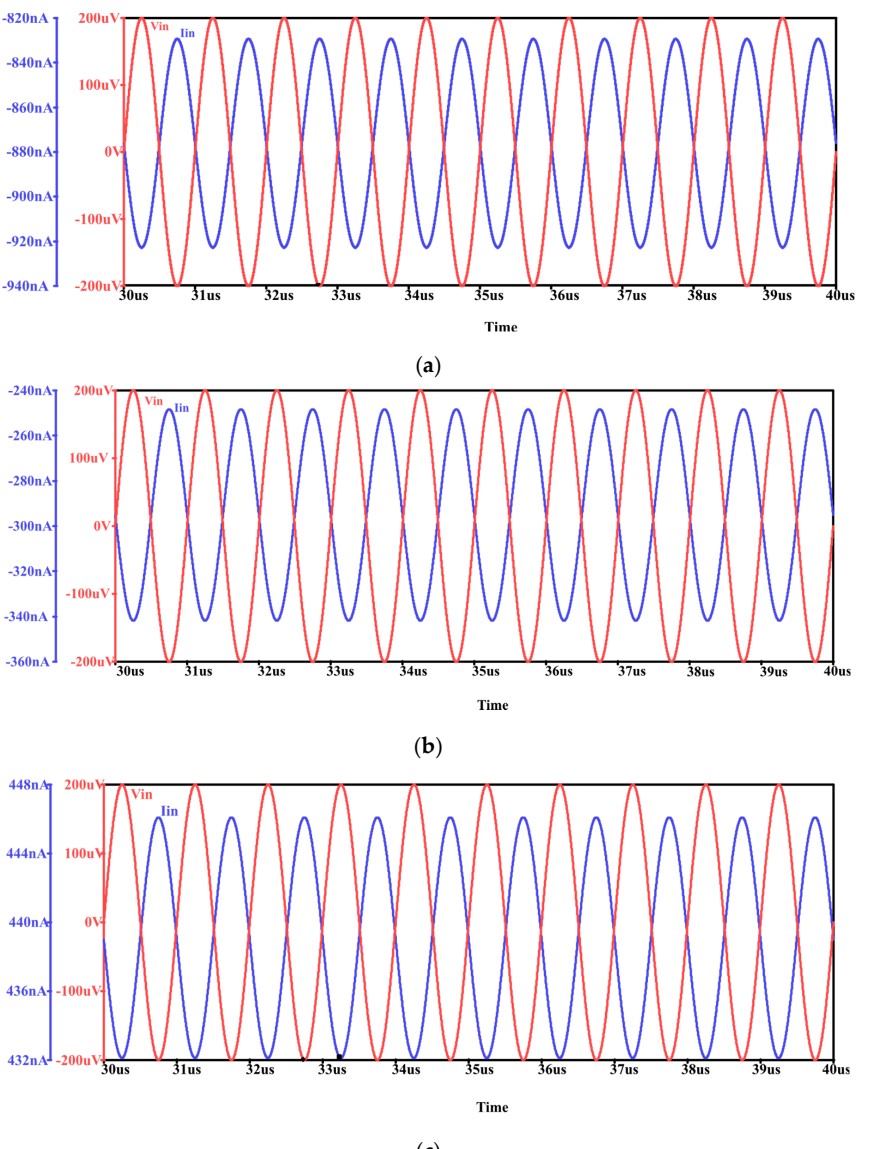

**Figure 11.** Time domain response of the proposed circuit (**a**) $V_C = -0.9$ V (**b**) $V_C = -0.45$ V, and (**c**) $V_C = 0$ V.

A comparison between the proposed circuit and other previously reported ones is presented in Table 4. As can be seen, the proposed circuit outperforms previous works in terms of reduced supply voltage, resistor-free structure, electronic tunability, reduced error, and simplicity.

**Table 4.** Comparison between proposed circuit and other reported works.

| Electronic Tuning | Max Error | Vdd–Vss | Power Consumption | Number of Transistors | Passive Elements | | ABB | Ref |
|---|---|---|---|---|---|---|---|---|
| | | | | | R | Floating C | | |
| Yes | 20% [1] | NA | NA | NA | 1 | No | 2OTA + CCII | [7] |
| Yes | NA | ±1.25 V | 4.05 Mw [2] | 99 | 0 | No | 3CCDDC | [8] |
| Yes | NA | ±1.5 V | NA | 104 | 0 | No | 4CFTA | [9] |
| Yes | NA | 2 V | 0.7 mW | 47 | 0 | Yes | VGC-CCII | [10] |
| Yes | NA | ±2.5 V | 4.98 mW | 70 | 0 | No | Do-CCII + 3CCCII | [11] |
| No | NA | ±1.5 V | 9.52 mW | 36 | 2 | No | 2DVCC | [12] |
| No | NA | ±0.75 V | 1.29 mW | 98 | 2 | No | 2MODVCC | [13] |
| Yes | 8.60% | ±0.75 V | 2.3 μW–6.34 μW | 76 | 0 | Yes | CCII + 4OTA | [14] |
| No | NA | NA | NA | NA | 2 | No | 2CCII | [15] |
| Yes | 10%1 | ±2 V | NA | 24 | 1 | No | DVCCTA | [16] |
| No | 7.60% | 1.3 V | 1.32 mW | 72 | 0 | yes | 2OTA | [17] |
| No | NA | 1.5 V | 240 μW | 34 | 0 | Yes | FVF | [18] |
| Yes | 8% | ±2.5 V | 0.565 mW | 104 | 0 | No | 4Gm | [19] |
| No | NA | ±0.45 V | 0.556 mW | 56 | 2 | No | 2DVCC | [20] |
| No | NA | 1.8 V | 5.72 μW | 11 | 0 | Yes | MOS | [21] |
| Yes | 0.56% | ±0.9 V | 2.234 mW–3.184 mW | 51 | 0 | No | VCII$^{\pm}$ + E-DVCC | Proposed |

[1] Calculated. [2] Power consumption for control current of 10 μA.

## 6. Conclusions

In this paper, we present a new resistor-free topology for implementing either a positive or negative floating capacitor multiplier with electronic tuning capability. One VCII$^{\pm}$, one E-DVCC, and a single grounded capacitor are used. Using an electronically tunable current gain stage between the X and Z terminals of the E-DVCC, various multiplication factors are achieved. The circuit enjoys low voltage operation and offers reduced error compared with other works. A non-ideal analysis is given by taking into account non-ideal gains and parasitic impedances of the used active elements. The proposed circuit is free from any restricting matching requirements. The application of the proposed circuit as a standard high pass filter is also given to verify its functionality.

**Author Contributions:** Editing, L.S., G.F. and V.S.; visualization, G.B., M.S. and L.S.; supervision, G.F. and V.S.; project administration, G.F. and V.S.; funding acquisition, G.F. and V.S. All authors have read and agreed to the published version of the manuscript.

**Funding:** This research has been partially funded by the European co-funded innovation project iRel4.0 ECSEL under grant agreement No. 876659.

**Institutional Review Board Statement:** Not applicable.

**Informed Consent Statement:** Not applicable.

**Data Availability Statement:** Data sharing not applicable. No new data were created or analyzed in this study. Data sharing is not applicable to this article.

**Conflicts of Interest:** The authors declare no conflict of interest.

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
