# Peer review of "New Resistor-Less Electronically Controllable ±C Simulator Employing VCII, DVCC, and a Grounded Capacitor"

_electronics, doi:10.3390/electronics11020286_

Round 1

Reviewer 1 Report

This paper aims at proposing a realization of electronically controllable positive and negative floating capacitor multiplier. Although the topic is interesting, I believe that there are some points that should be better explained by the authors:

  1. The authors could explain more clearly the advances in the proposed work in relation to their previous works of [2] and [22].
  2. When comparing (11) and (17a), it seems that the term related to Vx+ in Ix+ simply disappears. This should be better explained in the equations’ development.
  3. Besides, I believe that it should be written “?1 = ??+ = ??1?y” in (17a).
  4. The last paragraph of page 8 and the first sentence of page 9 are repeated in the text.
  5. How have the dimensions of the device been defined?
  6. Have the Spice simulations been validated with some experimental data and/or results from the literature?
  7. How have the parasitic impedances shown in Table 2 been extracted? All of them are resistances. Aren’t there any parasitic capacitance?
  8. What is the frequency operation range of the proposed circuit?
  9. The curves in fig. 8 should be better identified.
  10. In which condition, e.g. frequency, were the errors in Table 3 calculated?

Reviewer 2 Report

In Abstract, “Simulation results using SPICE program shows…” should be replaced by “Simulation results using SPICE program show…”.

In line 186, it is expressed that “Rx and Cx are parasitic parallel resistance and capacitance…”, but Cx is not illustrated in Figure 5. Moreover, in Figure 5, rx+ and rx- are shown and not Rx.

In page 8, the text “The transistor level implementation of an electronically tunable E-DVCC is shown in Figure 7. The difference between input voltages V1 and V2 is produced at low impedance X port utilizing negative feedback loop established by differential pairs M1-M2 and M3-M4 along with M5-M7 transistors as reported in [16]. In the previously reported works, DVCC is used while the gain between X and Z terminals is unity. Here we add a gain cell between X and Z terminals to electronically tune the current at Z terminal. The used gain cell is a variable gain current mirror previously reported in [24].”  is repeated twice.

The significance of quantities appearing in equations (2), (26a) and (26b) should be expressed.

In Fig. 8, it should be shown which curves are for magnitude and which are for phase.

In line 241, in “Figure10”, a space should be after the word “Figure”.

Round 2

Reviewer 1 Report

I believe that the points mentioned in the previous manuscript version were adequately addressed by the authors.